# Suppression of choroidal neovascularization and epithelial-mesenchymal transition in retinal pigmented epithelium by adeno-associated virus-mediated overexpression of CCN5 in mice

Sora Im[1‡], Jung Woo Han[2‡], Euy Jun Park[1], Ji Hong Bang[3], Hee Jeong Shin🆔[3], Hun Soo Chang[4], Kee Min Woo[5☯], Woo Jin Park🆔[1,5☯]*, Tae Kwann Park[2,3,6☯]*

1 College of Life Sciences, Gwangju Institute of Science and Technology, Gwangju, Korea, 2 Department of Ophthalmology, Soonchunhyang University, College of Medicine, Bucheon, Korea, 3 Department of Interdisciplinary Program in Biomedical Science, Soonchunhyang Graduate School, Bucheon Hospital, Bucheon, Korea, 4 Department of Anatomy and BK21 Four Project, College of Medicine, Soonchunhyang University, Cheonan, Korea, 5 Olives Biotherapeutics, Inc., Gwangju, Korea, 6 Laboratory for Translational Research on Retinal and Macular Degeneration, Soonchunhyang University Hospital Bucheon, Bucheon, Korea

☯ These authors contributed equally to this work.
‡ SI and JWH also contributed equally to this work.
* tkpark@schmc.ac.kr (TKP); woojinpark@icloud.com (WJP)

**Data Availability Statement:** All relevant data are within the paper and its Supporting Information files.

## Abstract

Choroidal neovascularization (CNV) is a defining characteristic feature of neovascular age-related macular degeneration (nAMD) that frequently results in irreversible vision loss. The current strategies for the treatment of nAMD are mainly based on neutralizing vascular endothelial growth factor (VEGF). However, anti-VEGF therapies are often associated with subretinal fibrosis that eventually leads to damages in macula. In this study, we tested whether an anti-fibrotic and anti-angiogenic protein CCN5 can potentially be an effective and safe therapeutic modality in a mouse model of CNV. Laser photocoagulation was utilized to induce CNV, which was followed by intravitreal injection of recombinant adeno-associated virus serotype 2 encoding CCN5 (rAAV2-CCN5). Our data demonstrated that rAAV2-CCN5, but not a control viral vector, rAAV2-VLP, prominently attenuated both CNV lesions and angiogenesis. Aflibercept, which was utilized as a positive control, exhibited similar effects on CNV lesions and angiogenesis in our experimental settings. Upon laser photocoagulation, retinal pigmented epithelium (RPE) cells underwent significant morphological changes including cellular enlargement and loss of hexagonality. rAAV2-CCN5 significantly normalized these morphological defects. Laser photocoagulation also led to fibrotic deformation in RPE cells through inducing epithelial-mesenchymal transition (EMT), which was completely blocked by rAAV2-CCN5. In a striking contrast, aflibercept as well as rAAV2-VLP failed to exhibit any effects on EMT. Collectively, this study suggest that CCN5 might provide a potential novel strategy for the treatment of nAMD with a capability to inhibit CNV and fibrosis simultaneously.

**Funding:** K.M.W was supported by grant from Business for Startup growth and technological development (TIPS Program, S2842182) funded by Korea Ministry of SMEs and Startups in 2020. W.J.P was supported by grant from the National Research Foundation of Korea (2022R1A4A200076711, 2022R1A2C1004256) funded by the Korean government (MSIT). MSIT is the abbreviation of Ministry of Science and ICT. The funder had no role in study design, data collection and analysis, decision to publish, or preparation of the manuscript.

**Competing interests:** K.M.W. and W.J.P share co-ownership of Olives Biotherapeutics. No potential conflicts of interest exist for other authors.

## Introduction

Age-related macular degeneration (AMD) is a common cause of irreversible central blindness in older people [1]. AMD is categorized into geographic atrophy (aAMD) or "dry" AMD and neovascular (n) AMD, or "wet" AMD. nAMD causes irreversible central vision loss attributable to choroidal neovascularization (CNV), subretinal hemorrhage, photoreceptor cell death, and subsequently permanent subretinal fibrovascular scarring [2]. nAMD accounts for 10~15% of all AMD cases. Current treatments are aiming at the reduction of vascular endothelial growth factor (VEGF) level via intravitreal injection of anti-VEGF antibodies or traps, which is given every 4~8 weeks [3, 4]. The need for frequent injections is economically and procedurally burdensome, and its patient compliance is generally poor [5]. Furthermore, sustained suppression of VEGF level is reported to be associated with adverse side-effects on retina [6, 7]. For example, one pilot study revealed that anti-VEGF drugs are likely to trigger the development of subretinal fibrosis [8]. Further, the CATT study showed that fibrovascular scarring was developed in 45.3% of all nAMD eyes within 2 years after the commencement of anti-VEGF therapies, which eventually led to permanent visual impairment. Fibrosis appears to be the major cause underlying the poor prognosis associated with the current anti-VEGF therapies. Therefore, it is urgent to develop therapeutic modalities that can control CNV and subretinal fibrosis simultaneously.

A biological process referred to as epithelial-mesenchymal transition (EMT) plays important roles in both physiological and pathological processes such as embryogenesis, tumor metastasis, and tissue fibrosis [9]. Under pathological conditions that cause CNV, retinal pigment epithelial (RPE) cells were shown to undergo EMT, which eventually leads to fibtoric deformations including loss of junctional integrity [10] and the morphological conversion to fibroblast-like cells [11, 12]. It was shown that EMT of RPE cells is associated with the progression of AMD [13–17].

CCN5 is a 29 kDa secreted protein that belongs to the cell communication network (CCN) family of matricellular proteins (CCN1–6) [18]. CCN5 is involved in diverse biological processes. For example, CCN5 inhibits the proliferation and mobilization of breast cancer and pancreatic adenocarcinoma cells, and the proliferation of vascular smooth muscle cells [19]. We previously showed that CCN5 can inhibit cardiac fibrosis and the associated ventricular remodeling in mice partly via inhibiting the TGF-β-SMAD signaling pathway [20]. More intriguingly, CCN5 can reverse the pre-established cardiac fibrosis partly via inducing myofibroblast-specific apoptosis [20]. Our group recently showed that CCN5 is expressed in ARPE-19 cells and mouse RPE, and that its expression level was significantly reduced under various pathological conditions including TGF-β treatment and knockout of *chemokine (C-C motif) ligand 2* (*ccl2*) [21]. Restoration of the CCN5 expression level was sufficient to normalize or reverse the pathology in ARPE-19 cells and RPE [21]. In addition to its anti-fibrotic activity, CCN5 was shown to possess anti-angiogenic activity as determined by an *in vitro* aortic ring assay [22]. Although the anti-angiogenic activity of CCN5 has not been vigorously studied, it may be attributable to the role of CCN5 in the suppression of the CCN2 expression, a potent pro-fibrotic and pro-angiogenic cytokine [23]. Therefore, it is compelling to test whether CCN5 is beneficial in *in vivo* models of nAMD. In this study, we utilized a recombinant adeno-associated virus serotype 2 encoding CCN5 (rAAV2-CCN5) to investigate the effects of CCN5 in the laser-induced CNV model.

## Materials and methods

### Animals

All animal experiments were conducted in accordance with the Guide for the Care and Use of Laboratory Animals and the Association for Research in Vision and Ophthalmology Statement

for the Use of Animals in Ophthalmic and Vision Research, and were approved by the Institutional Animal Care and Use Committee of Soonchunhyang University Hospital, Bucheon, Korea. All efforts were made to minimize the number of animals used and suffering.

Male C57BL/6 mice were purchased from Orient Bio Inc. (Seongnam, Korea) and DBL Inc. (Eumseong, Korea). All mice were maintained in a temperature- and humidity-controlled room with a 12/12-h light/dark cycle under standardized conditions, and had free access to food and water.

Anesthesia was induced via intraperitoneal injection of a mixture of 80 mg/kg Zoletil 50 (Virbac; Carros Cedex, France) and 20 mg/kg Rompun (Bayer Healthcare, Leverkusen, Germany) and pupil dilation was performed with a mixture of 0.5% (w/v) tropicamid and 0.5% (w/v) phenylephrin (Hanmi Pharm, Seoul, Korea) prior to all laser treatments and examinations.

## Laser-induced CNV in mice

Laser photocoagulation was performed as previously described [24]. Photocoagulation (200 μm spot diameter, 20 ms duration, 120 mW power, 4~5 spots/eye) was created using a diode ophthalmic laser system (Neodyminum-doped yttrium aluminium garnet [Nd:YAG], 532 nm; Topcon Medical Laser Systems, Livermore, CA) with mice under anesthesia and pupils dilated; the spots were produced at 12, 2, 5, 7, and 10 o'clock from the optic nerve with the laser focused on RPE. Development of a bubble was indicative of Bruch membrane disruption.

## Fundus fluorescent angiography (FFA)

FFA was performed as previously described with mice under anesthesia and pupils dilated [24]. We used a confocal, scanning laser ophthalmoscope (Heidelberg Retina Angiograph 2; Heidelberg Engineering, Heidelberg, Germany) to evaluate CNV status. FFA images were captured at 3 to 5 min after intraperitoneal injection of 150 μL 1% (w/v) fluorescein sodium (Fluorescite; Akorn, Lake Forest, IL). FFA images taken at the same time points were analyzed using ImageJ software (National Institutes of Health, Bethesda, MD) after manual selection of the maximal leakage areas. To measure the hyperfluorescent areas, FFA images were imported into ImageJ, where the maximal border of CNV lesions was manually outlined under digital magnification. The encompassed area measurement in pixels converted to $\mu m^2$ using the "scale" tool in ImageJ software. To measure the hyperfluorescent intensity, the fluorescence intensity within the maximum border of each CNV lesion was calculated using ImageJ software, areas where large vessels overlapped the hyperfluorescent area was excluded. Background fluorescent intensity was measured by defining an annulus area around the CNV lesion. The net fluorescent intensity above background was calculated by subtracting the calculated background value from the CNV hyperfluorescent intensity.

## Intravitreal recombinant AAV serotype 2 (rAAV2) injection

A rAAV2 construct harboring the human CCN5 gene under the control of the CMV promoter was generated and referred to as rAAV2-CCN5. A rAAV2 construct harboring the GFP, P2A, and human CCN5 gene under the control of the CMV promoter was generated and referred to as rAAV2-GFP-P2A-CCN5. rAAV2-CCN5 and rAAV2-virus-like-particle (VLP) viral vectors were produced and purified by Virovek (Hayward, CA). rAAV2-GFP-P2A-CCN5 vrial vector was produced and purified by Vigene Biosciences (Rockville, MD). Five days after laser photocoagulation and FFA, rAAV was intravitreally injected into both eyes with mice under anesthesia and pupils dilated. A puncture site was carefully created ~2 mm posterior to the

limbus using a 30-G beveled needle under an operating microscope. rAAV (1.2 μL) was injected through the puncture site employing a NanoFil syringe with a 34-G beveled needle (World Precision Instruments Inc., Sarasota, FL). Aflibercept was purchased from Bayer (Leverkusen, Germany) and 40 μg was injected per eye.

## Cells and cell culture

ARPE-19 cells were purchased from the American Type Culture Collection (ATCC, Manassas, VA) and cultured in DMEM/F-12 from Welgene (Gyeongsan, Korea) with 10% fetal bovine serum (HyClone, Logan, UT) and 1% penicillin-streptomycin (Gibco, Gaithersburg, MD) at 37˚C and 5% $CO_2$. At 80~90% cell confluency, the medium was replaced by that containing rAAV2-CCN5 at 1,000 multiplicity of infection (MOI) for 5 days.

## Western blot analysis

After 1 week of intravitreal rAAV injection, mice were sacrificed with a $CO_2$ chamber under deep anesthesia. The eyeballs were enucleated in PBS. The eyecups were made by removing connective tissues, cornea and optic nerve. Retina and RPE/choroid complex were separated from eyecup. To extract protein from tissues, the tissues were suspended with cold RIPA lysis buffer (1% NP-40, 50 mM Tris-HCl [pH7.4], 150 mM NaCl, and 10 mM NaF) containing protease inhibitor cocktail (PIC; Roche, Basel, Switzerland) and sonicated at an amplitude of 50% for 10 seconds with a cycle of 2 seconds on/1 seconds off using probe sonicator (SONICS, Newtown, CT) with 1/8" probe.

ARPE-19 cells were washed with PBS and scrapped. ARPE-19 cells lysates were obtained by solubilizing with cold RIPA lysis buffer containing PIC and sonication at an amplitude of 50% for 5 minutes with a cycle of 2 seconds on/1 seconds off.

Lysates were centrifuged at 13,000 rpm for 20 min at 4˚C. The protein concentration of the supernatant was quantified using Pierce BCA Protein Assay kit (Thermo Fisher Scientific, Waltham, MA). Equal amount of protein samples were mixed with protein 5x sample buffer (Elpis Biotech, Daejeon, Korea) and incubated for 30 min at 37˚C. The denatured samples were loaded for SDS-PAGE and transferred to polyvinylidene difluoride membranes (Merck, Kenilworth, NJ). The membranes were blocked with 5% (w/v) skim milk in Tris-buffered saline containing 0.1% (v/v) Tween 20 (TBS-T) for 1 h at room temperature (RT) and incubated with primary antibodies in 3% (w/v) skim milk in TBS-T overnight at 4˚C, washed with TBS-T and labeled with the appropriate secondary antibodies in TBS-T for 1 h at RT. They were washed then, immune complex was visualized using a Western Femto ECL kit (LPS solution, Daejeon, Korea) and ImageQuanti LAS 4000 Mini imager (GE Healthcare, Chicago, IL). Primary antibodies were anti-CCN5 (Genscript Biotech, Piscataway, NJ), anti-CCN5 (LSBio, Seattle, WA), anti-CCN2 (Abcam, Cambridge, UK), β-actin (Santa Cruz, Dallas, TX), and GAPDH. Secondary antibodies against mouse and rabbit (conjugated with horseradish peroxidase) were purchased from Invitrogen (Carlsbad, CA).

## Immunofluorescence analysis

Seven days after intravitreal rAAV injection, the mice were sacrificed with a $CO_2$ chamber under deep anesthesia and perfused with 4% (v/v) paraformaldehyde. The posterior eyecups were fixed in 4% (v/v) paraformaldehyde for 1 h at 4˚C and dehydrated in 30% (w/v) sucrose for 12–16 h at 4˚C. Neural retina was removed from RPE/choroid complex. RPE/choroid complex were washed sequentially with 70% (v/v) ethanol and PBST (PBS containing 0.1% (v/v) Triton X-100), blocked with 5% (v/v) donkey serum for 3 h, incubated with primary antibodies in 5% (v/v) donkey serum overnight at 4˚C, washed with PBST, and then incubated with

secondary antibodies and Hoechst 33342 (Invitrogen, Carlsbad, CA) in PBS for 2 h at RT. They were then washed, mounted on slide glass using mounting medium (Dako, Santa Clara, CA), and imaged under a microscope fitted with a Leica confocal camera (DMI8; Leica Camera, Wetzlar, Germany). Primary antibodies were anti-CD31 (BD Biosciences, San Jose, CA), anti-fibronectin (Abcam, Cambridge, UK), anti-vimentin (Abcam, Cambridge, UK), anti-α-smooth muscle actin (α-SMA; Sigma-Aldrich, St. Louis, MO), and anti-zonula occludens-1 (ZO-1; Invitrogen, Carlsbad, CA). Secondary antibodies against rat (labelled with Alexa Fluor 594), rabbit (labelled with Alexa Fluor 488), mouse (labelled with Alexa Fluor 647) IgG were obtained from Invitrogen (Carlsbad, CA).

For immunohistochemistry, the posterior eyecups were embedded with optimal frozen section compound (Leica, Wetzlar, Germany), and they were stored at -80˚C until use. Tissues were sectioned at 10 μm in thickness and mounted onto micro slide glass (Matsunami, Osaka, Japan). Slides were dried for 30 min at 50˚C and washed with PBST for 15 min. The slides were blocked with 5% (v/v) normal goat serum (Abcam, Cambridge, UK) in PBST for 1 h, incubated with primary antibodies in 5% (v/v) normal goat serum for overnight at 4˚C, washed with PBST, and then incubated with secondary antibodies in PBS for 2 h at RT, washed, incubated with Hoechst 33342 in PBS for 15 min at RT. The tissue sections were then washed and covered with coverslips using mounting medium and imaged under a Zeiss confocal microscope (LSM880 NLO; Carl Zeiss, Oberkochen, Germany) and Olympus confocal (FV3000RS; Olympus, Shinjuku, Tokyo, Japan). Additional primary antibodies were anti-GFP (Abcam, Cambridge, UK) and anti-glial fibrillary acidic protein (GFAP; Abcam. Cambridge, UK).

### Cell counting

For cell counting, the flat mounts of RPE/choroid complex were stained with antibody against ZO-1. Voronoi diagram was generated using Rhinoceros algorithm (Rhinoceros ver. 6.0; Roberts McNeel & Associates, Seattle, WA) and ImageJ software, as previously described [25]. 12 days after CNV induction, the number of RPE cells were counted those are located within a 200 μm diameter circle centered on the CNV-lesion. Also, the area within the circles to evaluate the changes of RPE cell size and hexagonal morphology. The area of 200 μm diameter circle was divided by the number of RPE cells to calculated the mean RPE cell areas.

### Image analysis and Statistical analysis

All experiments were repeated independently at least three times. Immunofluorescenc images was imported into ImageJ software, and the fluorescent intensity was calculated by defining an annulus area around the CNV lesion. Statistical significance of difference was analyzed by one-way analysis of variance (ANOVA), two-way ANOVA, or unpaired Student's two-tailed t-test using GraphPad Prism 9 software (GraphPad Software, San Diego, CA).; significant differences are indicated by a single asterisk (*, $p < 0.05$), a double asterisk (**, $p < 0.01$), or a triple asterisk (***, $p < 0.001$). Each line represents median, each dot represents mean, and each bar represents the S.D. for parametric data.

## Results

### Transduction efficiency of rAAV2-CCN5

The experimental viral vector, rAAV2-CCN5 and rAAV2-GFP-P2A-CCN5, were prepared for identification of transduction efficiency (Fig 1A, S1 Fig). To induce CNV in mouse eyes, laser photocoagulation was performed. At 5 days after laser-induced CNV, rAAV2-CCN5 was

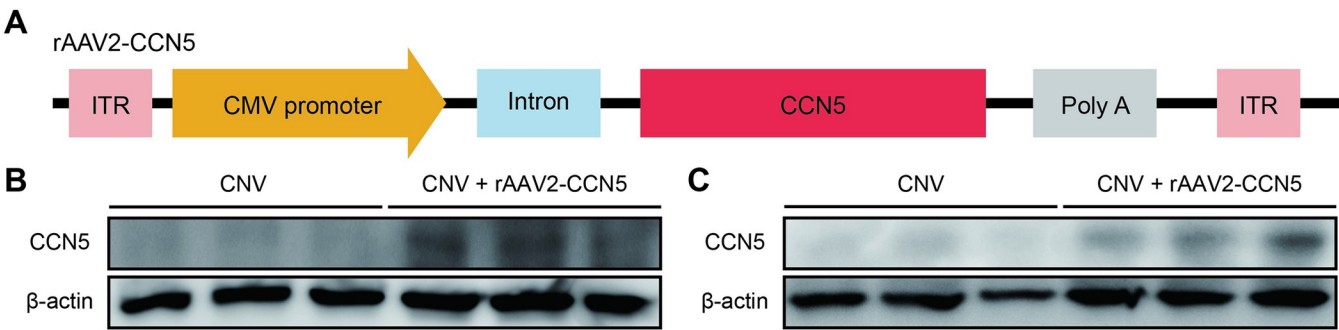

**Fig 1. Design and validation of rAAV2-CCN5.** (A). A schematic representation of rAAV2-CCN5. ITR, inverse terminal repeat. (B, C). CCN5 expression was determined via western blotting from retina and RPE/choroid complex. (n = 3~6 mice per group).

administered via intravitreal injection. One week after injection, western blotting was carried out to determine the transduction efficiency of intravitreally-delivered rAAVs. The results showed that the levels of CCN5 expression were upregulated in retina (Fig 1B, S1 Fig) and RPE/choroid complex (Fig 1C, S1 Fig) of rAAV2-CCN5 and rAAV2-GFP-P2A-CCN5 injection group. Also, CCN5 expression was observed in ARPE-19 cells that were infected with rAAV2-CCN5 (S2 Fig). The transduction efficiencies of intravitreally-delivered rAAV were also determined by evaluating GFP expression on transverse retinal sections. GFP expression was detectable from ganglion cell to RPE layer after treatment (S1 Fig). These data indicate CCN5 gene was efficiently transduced into retina and RPE.

## Effects of rAAV2-CCN5 on CNV leakage

To study the effects of CCN5 on CNV leakage, a recombinant AAV vector that drives the expression of human CCN5 under the control of CMV promoter, rAAV2-CCN5 (Fig 1A), was generated. Laser photocoagulation was performed to induce CNV in mouse eyes. At five days after laser photocoagulation, FFA was performed to examine the vascular leakage and the formation of new vessels. Well-demarcated hyperfluorescent spots were observed in all laser photocoagulated eyes, which indicates that CNV lesions were well established in all groups. rAAV2-CCN5 or a control viral vector, rAAV2-VLP, was intravitreally administered at this time point. Aflibercept, one of the most commonly used anti-VEGF drugs, was utilized for comparison in this and all the following experiments. FFA was performed again at seven days following the administration of the viral vectors or aflibercept to examine the phenotypic consequences. The experimental scheme is shown in Fig 2A. Representative FFA images are shown in Fig 2B, and the hyperfluorescent area and intensity were quantified and plotted in Fig 2C and 2D, respectively. While rAAV2-VLP had no effects, rAAV2-CCN5 significantly reduced both the size and intensity of hyperfluorescence. Aflibercept, served as a postivie control, also effectively reduced the size and intensity of hyperfluorescence. The effects of rAAV2-CCN5 and aflibercept were statistically indistinguishable. These results indicate that CCN5 can inhibit CNV leakage.

## Effects of rAAV2-CCN5 on CNV

We then evaluated the effects of rAAV2-CCN5 on CNV. At the end of experiments shown Fig 2A, eyes were harvested and flat mounts of RPE/choroid complex were prepared. Immunostaining of the flat mounts with an antibody against CD31, a marker for vascular endothelial cells, was performed to visualize CNV area (Fig 3). Representative images are shown in Fig

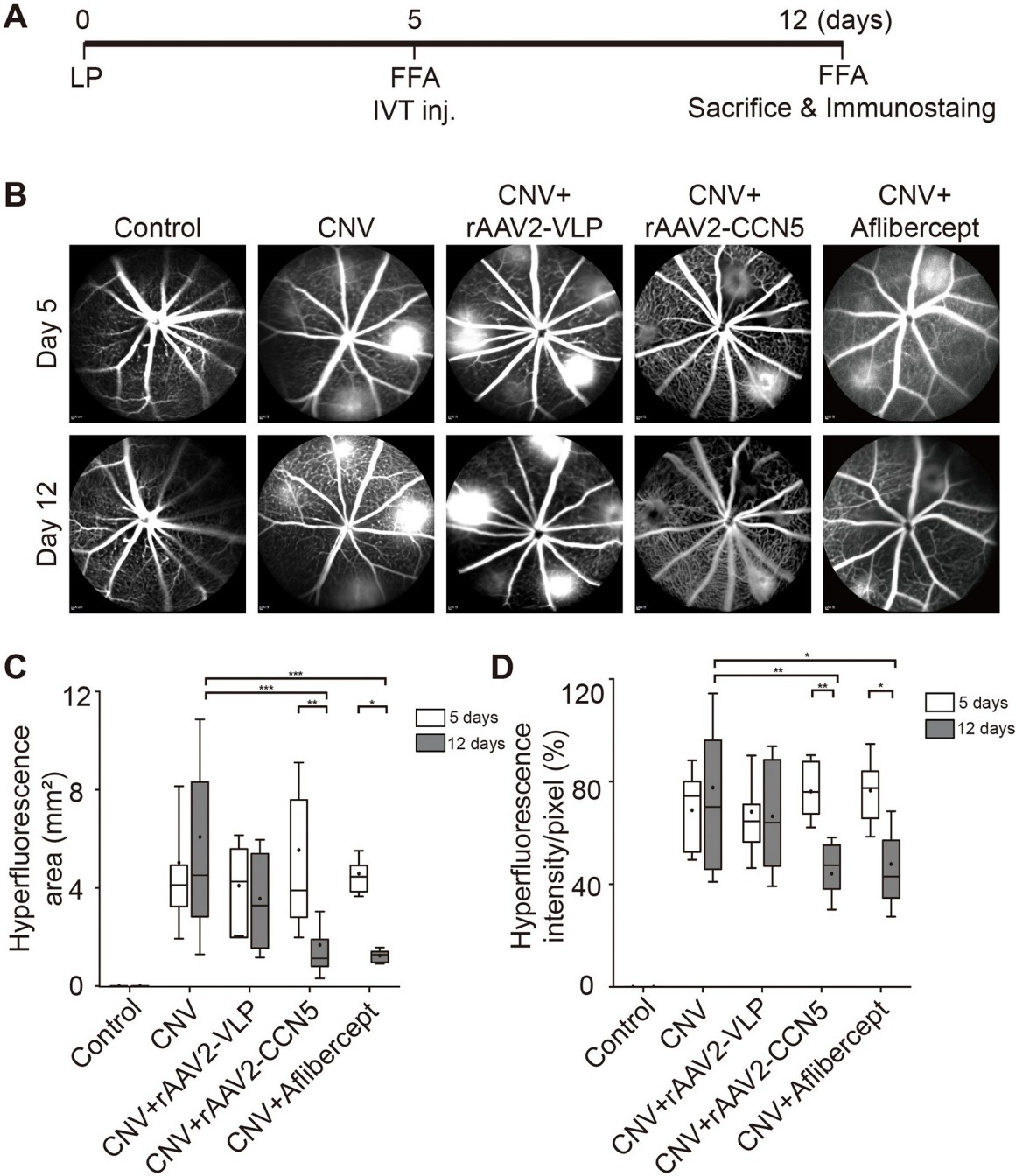

**Fig 2. FFA analysis of laser photocoagulation-induced CNV.** (A). An experimental scheme. Laser photocoagulation (LP) was performed on day 0. FFA was then performed on day 5, and followed by intravitreal (IVT) administration of rAAVs or aflibercept. Seven days post-injection, FFA was performed again and then animals were sacrificed. (B). Representative FFA images of CNV lesions captured on days 5 and 12 after LP are shown. (C, D). Quantitative analysis of FFA images in terms of hyperfluorescence areas and intensities, respectively. Bars show the mean ± SD. (n = 13~21 spots per group; two-way ANOVA; $^*p < 0.05$, $^{**}p < 0.01$, $^{***}p < 0.001$).

3A~3O. The CD31-positive CNV area was quantitated using ImageJ software and plotted (Fig 3P). Laser photocoagulation prominently increased the CD31-positive area, which was significantly blocked by rAAV2-CCN5 or aflibercept, but not by rAAV2-VLP. These data indicate that CCN5 can inhibit CNV.

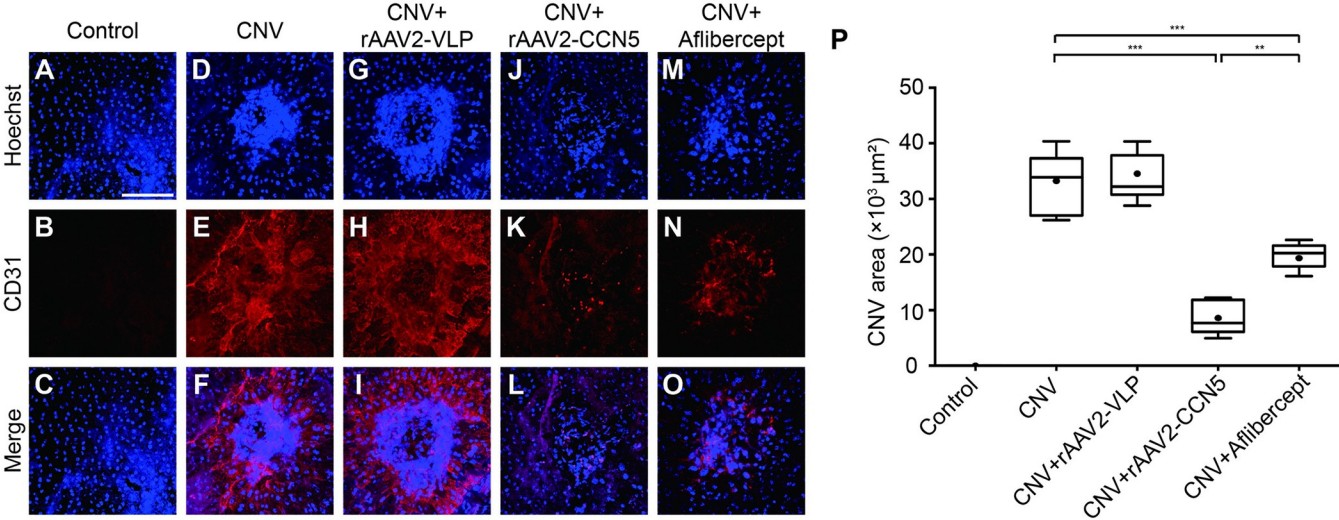

**Fig 3. Assessment of CNV by immunostaining for CD31.** (A~O). Flat mounts of RPE/choroid complex were immunostained for CD31. Hoechst dye was used for staining of nuclei. Representative images are shown. Scale bar, 100 µm. (P). CD31-positive areas were quantified using ImageJ algorithm and plotted. Bars show the mean ± SD. (n = 6 spots per group; one-way ANOVA; $^{**}P < 0.01$, $^{***}p < 0.001$).

## Effects of rAAV2-CCN5 on retinal gliosis

Reactive gliosis of retinal glial cells is often associated with nAMD. To investigate the effects of rAAV2-CCN5 on retinal gliosis, we immunostained transverse retinal sections with an antibody against GFAP, a marker for retinal gliosis (Fig 4). The GFAP expression was significantly increased throughout retina upon laser photocoagulation (Fig 4G), which was significantly

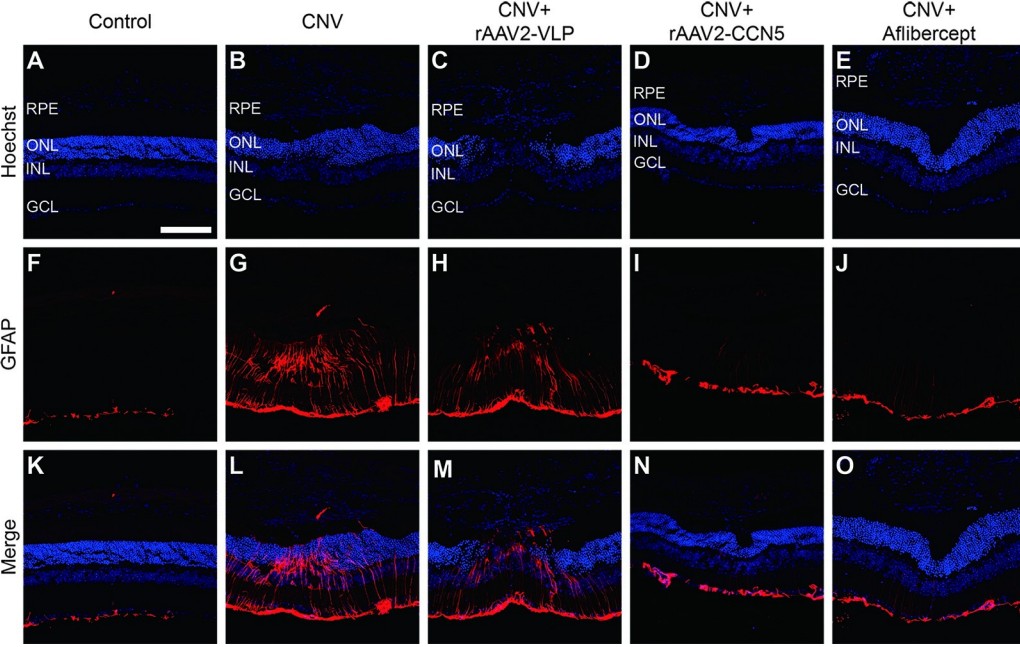

**Fig 4. Assessment of retinal gliosis by immunostaining for GFAP.** (A-O). Transverse retinal sections were immunostained for GFAP. Hoechst dye was used for staining of nuclei. RPE, retinal pigmeted epithelium; ONL, outer nuclear layer; INL, inner nuclear layer; GCL, ganglion cell layer. Scale bar, 100 µm.

inhibited by rAAV2-CCN5 or aflibercept, but not by rAAV2-VLP (Fig 4H–4J, respectively). The effects of rAAV2-CCN5 and aflibercept were indistinguishable. These data indicate that CCN5 can inhibit retinal gliosis.

## Effects of rAAV2-CCN5 on cell morphology

To visualize the morphology and integrity of RPE cells, flat mounts of RPE/choroid complex were stained with an antibody against ZO-1, a tight junction protein. Individual RPE cells were clearly demarcated by the ZO-1 immunostaining as shown in representative confocal images (Fig 5A). The confocal images were further converted to extracted images by manual tracing (Fig 5B). We found that the hexagonality and homogeneity of RPE cells were severely disrupted in the laser photocoagulated area. However, these morphological changes were significantly normalized by rAAV2-CCN5 or aflibercept, but not by rAAV2-VLP.

To further quantitate the changes in cell size, corresponding Voronoi diagrams were generated from the extracted images using a computer algorithm (Fig 5C). Circles with 200 μm in diameter were then virtually placed on the center of laser photocoagulated areas, and the numbers and sizes of RPE cells within the circles were quantitated. RPE cells were significantly enlarged upon laser photocoagulation. Accordingly, cell numbers were significantly reduced within the injured areas. rAAV2-CCN5 or aflibercept, but not rAAV2-VLP, significantly normalized these morphological abnormalities as illustrated by the decrease in cell size (Fig 5D) and the increase in cell number (Fig 5E). These data indicate that CCN5 can normalize morphological defects induced by laser photocoagulation.

## Effects of rAAV2-CCN5 on EMT

Laser photocoagulation was shown to induce subretinal fibrosis that is associated with EMT of RPE cells. We previously showed that CCN5 inhibits TGF-β-induced EMT in ARPE-19 cells. Theferefore, we tested whether CCN5 can inhibit laser photocoagulation-induced EMT in RPE cells. Flat mounts of RPE/choroid complex were stained with antibodies against α-SMA, vimentin, and fibronectin, markers for EMT (Fig 6A–6C, respectively). Expression levels of these marker proteins were dramatically increased by laser photocoagulation, which indicates that RPE cells underwent EMT upon laser photocoagulation. This deformation was significantly inhibited by rAAV2-CCN5, but not by aflibercept nor rAAV2-VLP. Quantification of the expression levels of α-SMA, vimentin, and fibronectin was performed and plotted (Fig 6D–6F, respectively). These data indicate that CCN5 can inhibit EMT in RPE cells.

## Conclusion

Collectively, our data demonstrate that rAAV2-CCN5 can inhibit CNV and EMT in RPE cells that are induced by laser photocoagulation in mice. As expected, aflibercept can inhibit CNV under the same experimental condition. However, aflibercept failed to inhibit EMT in contrast to rAAV2-CCN5. Therefore, we suggest that rAAV2-CCN5 can be a safer yet efficient therapeutic modality for nAMD.

## Discussion

Subretinal fibrosis has been regarded as a key pathological event associated with CNV. Recent advances have allowed clinicians to treat CNV-related diseases by intravitreal administration of anti-VEGF antibodies or traps that lead to CNV regression, reductions in subretinal and choroidal exudations, and visual improvement. However, one alarming issue is that anti-VEGF drugs frequently induce a fibrotic response in RPE, which eventually leads to damages

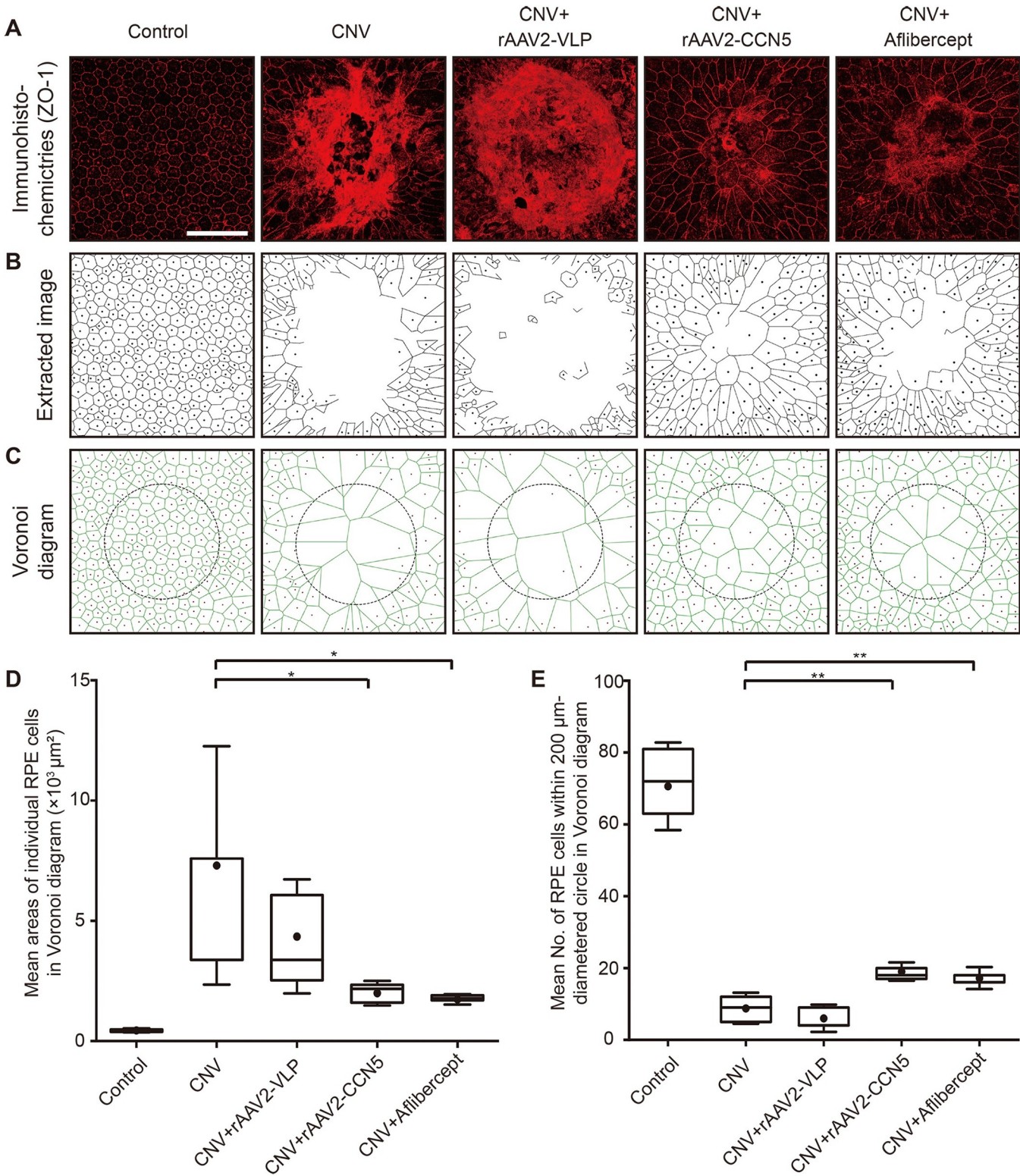

**Fig 5. Assessment of RPE cell morphology by a Voronoi diagram.** (A). Flat mounts of RPE/choroid complex were immunostained for ZO-1. Representative images are shown. Scale bar, 100 μm. (B). Extracted images were drawn to demarcate cell boundaries. (C). Voronoi diagrams were virtually generated from the extracted images using computer algorithm. (D). The mean sizes and (E) the mean numbers of RPE cells within the circles of 200 μm in diameter are calculated and plotted. Bars show the mean ± SD. (n = 5 spots per group; one-way ANOVA or unpaired Student's two-tailed t-test; *p < 0.05, **p < 0.01).

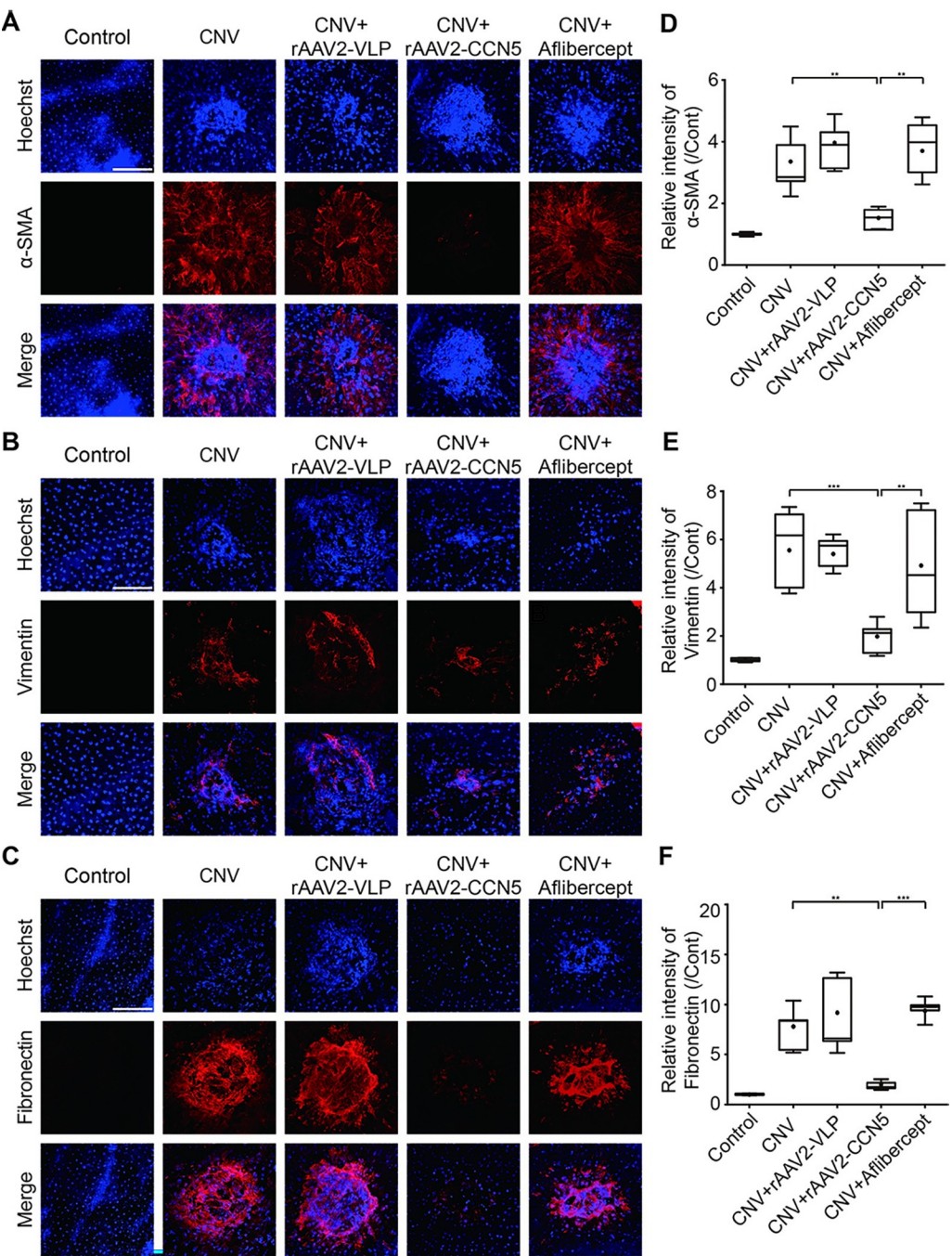

**Fig 6. Assessment of EMT by immunostaining for α-SMA, vimentin, and fibronectin.** (A-C). Flat mounts of RPE/ choroid complex were immunostained for α-SMA, vimentin, and fibronectin. Representative images are shown. Scale bar, 100 μm. (D-F). α-SMA-, vimentin-, and fibronectin-positive areas were captured and their intensities were calculated using Leica confocal software and plotted. Bars show the mean ± SD. (D, n = 6 spots per group; E, n = 7 spots per group; F, n = 5 spots per group; one-way ANOVA; **$p < 0.01$, ***$p < 0.001$).

in macula [26]. Therefore, new therapeutic modalities that can attenuate both CNV and sub-retinal fibrosis are urgently needed.

The matricellular proteins of the CCN family have recently been described [27]. These proteins play roles in diverse normal and pathological processes including cell proliferation,

migration, tumorigenesis, fibrosis, and angiogenesis [28]. CCN5 was shown to inhibit tumor progression at least in part by inhibiting the expression of genes involved in the TGF-β signaling cascade [29]. We previously showed that CCN5 inhibits cardiac fibrosis partly through inhibition of fibroblast-to-myofibroblast trans-differentiation [20]. We further showed that CCN5 inhibits fibrotic deformation of ARPE-19 cells under various pathological conditions including TGF-β treatment and knockout of *ccl2*. More strikingly, CCN5 can reverse pre-established fibrotic deformation of the ARPE-19 cells, as shown by the re-appearance of tight junction proteins and the recovery of RPE function including phagocytosis [21]. Molecular mechanisms underlying the anti-fibrotic activity of CCN5 are largely unknown. However, all the preliminary data indicate that CCN5 might act as a transcriptional cofactor and drives expression of genes involved in mesenchymal-epithelial transition (MET) in RPE.

In addition to its anti-fibrotic activity, CCN5 was also shown to possess anti-angiogenic activity as assessed by an aortic ring assay [22]. Angiogenesis begins with proliferation of endothelial cells, which is induced by VEGF, and followed by tube formation and maturation, which is mediated by various cytokines including CCN2, connective tissue growth factor (CTGF) [30, 31]. It is of note that CCN5 antagonizes CCN2 during cardiac fibrosis [23]. Thus, we hypothesized that CCN5 can exert its anti-angiogenic activity via suppression of the CCN2 expression in RPE. Further in-depth studies are warranted.

rAAV is currently the most frequently used vehicle for the *in vivo* delivery of therapeutic genes. The recent clinical success of rAAV2-RPE65 (Luxturna) for the treatment of Leber congenital amaurosis suggest that AAV vectors may also be successfully employed to treat other ocular diseases [32–34]. We previously demonstrated an enhanced transduction efficiency of intravitreally administered AAVs into various retinal cells after retinal laser photocoagulation, implying that the transduction of rAAV vectors may be enhanced by the pathologic state of retina [35, 36]. In this study, we were able to induce the expression of CCN5 in RPE cells by intravitreal delivery of rAAV2-CCN5. Laser photocoagulation appeared to enhance the transduction of rAAV2-CCN5 to retina and RPE, paticularly in the injured area.

Gliosis is activated by many pathogenic stimuli, including degenerative processes, inflammation, mechanical injury, and aging [37, 38]. The extent of physiological, biochemical, and morphological changes of gliosis depends on the severity of damages. Gliosis can damage neurons and vessels directly and indirectly, promoting neurodegeneration in patients with chronic retinal diseases such as AMD [39]. However, little is known about the effects of anti-VEGF drugs on retinal gliosis. We found that both rAAV2-CCN5 and aflibercept reduced retinal gliosis in the laser photocoagulated retinas. Although the molecular mechanisms underlying these beneficial effects are unclear yet, these data further illustrate the potential clinical benefit of rAAV2-CCN5.

RPE cells play fundamental roles in retinal structure and visual function. RPE cell dysfunction is an important feature of the pathophysiology of several macular diseases including AMD [40]. Normally, RPE cells exhibit a mature epithelial phenotype and are quiescent and in mutual contact. Subretinal fibrosis is attributable to the extensive tissue repair required after CNV [41]. In nAMD, the RPE layer and the Bruch membrane are penetrated mechanically or chemically as CNV proceeds [40, 42]. The RPE cells then lose cell-cell contact, proliferate, and undergo EMT that is involved in the fibrotic changes in AMD [41]. Aflibercept was able to inhibit CNV but not EMT in RPE cells upon laser photocoagulation. On a contrary, rAAV2-CCN5 inhibited CNV and protected RPE cells from undergoing EMT as well.

In summary, we found that rAAV2-CCN5 reduced angiogenesis in the laser photocoagulation-induced CNV lesions, and decreased the overall extent of CNV. In addition, CCN5 significantly reduced EMT in the CNV lesions, suggesting that CCN5 might suppress the post-EMT subretinal fibrosis. Further, CCN5 significantly normalized RPE cell morphology. Collectively,

our data suggest that rAAV2-CCN5 provides a platform for the development of safer yet effective therapeutics for the treatment of nAMD.

## Supporting information

**S1 Fig. Design and validation of rAAV2-GFP-P2A-CCN5.** (A). A schematic representation of rAAV2-GFP-P2A-CCN5. (B). Immunostaining of GFP in transverse retinal sections. GFP expression was detectable from ganglion cell to RPE layer. Scale bar, 100 μm. (C, D). CCN5 expression was determined via western blotting from retina and RPE/choroid complex. (n = 3~4 mice per group).
(TIF)

**S2 Fig. Transduction efficiency of rAAV2-CCN5 in ARPE-19 cells.** ARPE-19 cells were infected with rAAV2-CCN5 at MOI of 1,000 for 5 days. CCN5 expression was determined via western blotting from whole cells lysates. (n = 4 per group).
(TIF)

**S1 Raw images.**
(PDF)

## Acknowledgments

We thanks to Dr. H. Kim (GIST Central Research Facility) for technical assistance of the confocal microscopy.

## Author Contributions

**Conceptualization:** Kee Min Woo, Woo Jin Park, Tae Kwann Park.

**Funding acquisition:** Kee Min Woo, Woo Jin Park.

**Investigation:** Sora Im, Jung Woo Han, Ji Hong Bang, Tae Kwann Park.

**Methodology:** Sora Im, Euy Jun Park, Hee Jeong Shin.

**Project administration:** Woo Jin Park.

**Writing – original draft:** Sora Im, Jung Woo Han.

**Writing – review & editing:** Hun Soo Chang, Woo Jin Park, Tae Kwann Park.

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
