## [Decision Letter · Decision Letter 0]

28 Mar 2022

PONE-D-21-40491Suppression of choroidal neovascularization and epithelial-mesenchymal transition in retinal pigmented epithelium by adeno-associated virus-mediated overexpression of CCN5 in micePLOS ONE

Dear Dr. Park,

Thank you for submitting your manuscript to PLOS ONE. After careful consideration, we feel that it has merit but does not fully meet PLOS ONE’s publication criteria as it currently stands. Therefore, we invite you to submit a revised version of the manuscript that addresses the points raised during the review process. Please revise the manuscript as per the reviewer comments for further consideration.

We look forward to receiving your revised manuscript.

Kind regards,

Ravirajsinh Jadeja, Ph.D

Academic Editor

PLOS ONE

Journal Requirements:

2. Thank you for submitting the above manuscript to PLOS ONE. During our internal evaluation of the manuscript, we found significant text overlap between your submission and the following previously published works, some of which you are an author.

- https://pubmed.ncbi.nlm.nih.gov/33063711/

- https://journals.plos.org/plosone/article?id=10.1371%2Fjournal.pone.0132643

- https://link.springer.com/article/10.1134/S000629791809002X

- https://iovs.arvojournals.org/article.aspx?articleid=2532036&resultClick=1

Please revise the manuscript to rephrase the duplicated text, cite your sources, and provide details as to how the current manuscript advances on previous work. Please note that further consideration is dependent on the submission of a manuscript that addresses these concerns about the overlap in text with published work.

3. To comply with PLOS ONE submissions requirements, in your Methods section, please provide additional information regarding the experiments involving animals and ensure you have included details (including details regarding chemical concentrations) on (1) methods of sacrifice, (2) methods of anesthesia and/or analgesia, and (3) efforts to alleviate suffering.

Reviewers' comments:

Reviewer's Responses to Questions

**Comments to the Author**

1. Is the manuscript technically sound, and do the data support the conclusions?

Reviewer #1: No

Reviewer #2: Yes

2. Has the statistical analysis been performed appropriately and rigorously? 

Reviewer #1: Yes

Reviewer #2: Yes

3. Have the authors made all data underlying the findings in their manuscript fully available?

Reviewer #1: Yes

Reviewer #2: Yes

4. Is the manuscript presented in an intelligible fashion and written in standard English?

Reviewer #1: Yes

Reviewer #2: Yes

5. Review Comments to the Author

Reviewer #1: This study evaluated the dual effect of CCN5 as an anti-fibrotic and an anti-angiogenic protein in neovascular age-related macular degeneration. This is a well written and mostly comprehensive. The manuscript would be strengthened if the following issues are elaborated on in detail.

1. It is well known that the humanized VEGF antibody, Bevacizumab, is not capable of inhibiting the development of laser induced CNV in mice. Thus, it is very weird that Bevacizumab was used as a positive control and also it is working in this manuscript. Authors should elaborate on this critical issue.

2. In Figure 4, IHC staining with GFAP Ab does not clearly show active gliosis in the CNV and CNV+rAAV2-VLP groups. Thus, it would be difficult to draw results from these images.

3. In Figure 5, extracted images of hexagonal shape of RPE were acquired from IHC images with ZO-1. However, the ZO-1 signal is too weak to extract images which can affect the reliability of quantification of RPE cells. It would be better to elaborate on this issue.

Reviewer #2: This is a very well-designed study experimenting the anti-angiogenic and anti-fibrotic efficacy of AAV-CCN5 in mouse laser0induced CNV model. I have several comments as follows.

1. It will be also good if the transduction efficiency is presented in vitro also.

2. In fig 1. authors showed transduction efficiency by western blot. It will be also good if they show the immunofluorescence of the retinal section to show the retinal cells that expresses CCN5 after ITV injection of AAV-CCN5.

3. bevacizumab is known to only weakly bind to murine VEGF. Thus, it is inappropriate as a positive control. If authors want to present a positive control, aflibercept will be a better choice.

4. In the CNV model, recent data suggests that the active stage is between 2 to 5 days after laser treatment. Therefore, I am worrying the IVT administration of AAV or bevacizumab at 5 days is a little bit late and IVT injection at 0 or 1 days would be more appropriate to evaluate the anti-angiogenic efficacy on CNV.

6. PLOS authors have the option to publish the peer review history of their article (what does this mean?). If published, this will include your full peer review and any attached files.

Reviewer #1: No

Reviewer #2: No

---

## [Author Response · Author response to Decision Letter 0]

19 May 2022

Reviewer #1: This study evaluated the dual effect of CCN5 as an anti-fibrotic and an anti-angiogenic protein in neovascular age-related macular degeneration. This is a well written and mostly comprehensive. The manuscript would be strengthened if the following issues are elaborated on in detail.

1. It is well known that the humanized VEGF antibody, Bevacizumab, is not capable of inhibiting the development of laser induced CNV in mice. Thus, it is very weird that Bevacizumab was used as a positive control and also it is working in this manuscript. Authors should elaborate on this critical issue.

- > We appreciate this comment. As pointed out, the cross-reactivity of Bevacizumab to mouse VEGF is quite low. However, the amounts of bevacizumab we (and others) routinely use appears to be high enough to elicit anti-CNV activity in mice. As suggested, we performed the experiments all over again with aflibercept, which is known to have similar reactivity to human and mouse VEGF. As shown in Fig2~Fig6, aflibercept served as a nice positive control. 

2. In Figure 4, IHC staining with GFAP Ab does not clearly show active gliosis in the CNV and CNV+rAAV2-VLP groups. Thus, it would be difficult to draw results from these images.

- > We replaced the images to show active gliosis in the CNV more clearly as Fig4. 

3. In Figure 5, extracted images of hexagonal shape of RPE were acquired from IHC images with ZO-1. However, the ZO-1 signal is too weak to extract images which can affect the reliability of quantification of RPE cells. It would be better to elaborate on this issue.

- > Although the ZO-1 signal appears to be weak for automatic image extraction, the computer software successfully extracted the Voronoi diagram. To confirm this process, we repeated the image extraction by hand (Extracted image). These two images have no significant differences in the size and shapes of the cells. We replaced the figures with a more sharp images in Fig5.

Reviewer #2: This is a very well-designed study experimenting the anti-angiogenic and anti-fibrotic efficacy of AAV-CCN5 in mouse laser0induced CNV model. I have several comments as follows.

1. It will be also good if the transduction efficiency is presented in vitro also.

-> Thank you for the comment. Once we get recombinant AAVs, we routinely check the integrity of the viruses using alkaline gel electrophoresis and western blotting. We added one of the such western blotting data in S2 Fig.

2. In fig 1. authors showed transduction efficiency by western blot. It will be also good if they show the immunofluorescence of the retinal section to show the retinal cells that expresses CCN5 after ITV injection of AAV-CCN5.

-> We appreciate this comment. We previously showed that, in combination with laser photocoagulation, intravitreally injected AAV2 can be delivered to entire retina including RPE, photoreceptors, Muller cells, inner nuclear layer cells, and retinal ganglion cells (Human Gene Therapy Methods. 2014. 25:83-91, Mol Ther Nucleic Acids 2017. 8, 26-35). Thus, we routinely and successfully perform gene delivery utilizing IVT injection and laser photocoagulation. Moreover, it is of note that CCN5 is a secreted protein that can diffuse out to neighboring cells from the transfected cells, which is an obvious advantage as a gene therapy modality. Nevertheless, to further address this issue, we utilized rAAV2-CMV-GFP-P2A-CCN5 that expresses GFP and CCN5 simultaneously because there are no good anti-CCN5 antibodies suitable for IHC in tissue sections. The results show that GFP expression was detectable throughout the retinal layers from retinal ganglion cells to RPE cells as shown in S1 Fig. We also carried out western blotting showing that CCN5 expression was observed in retina and RPE/choroid complex. Fortunately, there are several anti-CCN5 antibodies are available suitable for western blotting.

3. VEGF. Thus, it is inappropriate as a positive control. If authors want to present a positive control, aflibercept will be a better choice.

-> This issue was also raised by the reviewer 1. See the Q1. We repeated all the experiments with aflibercept.

4. In the CNV model, recent data suggests that the active stage is between 2 to 5 days after laser treatment. Therefore, I am worrying the IVT administration of AAV or bevacizumab at 5 days is a little bit late and IVT injection at 0 or 1 days would be more appropriate to evaluate the anti-angiogenic efficacy on CNV.

- > We appreciate this comment. We previously showed that hyperfluorescence area and intensity were most strongly observed on day 5 after laser treatment. (Cell Commun Signal .2019 Jun 14;17(1):64). Expression of VEGF-related proteins in laser-induced CNV were also upregulated and peaked on day 5 after laser treatment. We wanted to show that CNV was induced at day 5 successfully among all the groups before rAAV or aflibercept were treated. For these reasons, the IVT administration was performed on the day 5.

---

## [Editor Report · Decision Letter 1]

1 Jun 2022

Suppression of choroidal neovascularization and epithelial-mesenchymal transition in retinal pigmented epithelium by adeno-associated virus-mediated overexpression of CCN5 in mice

PONE-D-21-40491R1

Dear Dr. Park,

We’re pleased to inform you that your manuscript has been judged scientifically suitable for publication and will be formally accepted for publication once it meets all outstanding technical requirements.

Kind regards,

Ravirajsinh Jadeja, Ph.D

Academic Editor

PLOS ONE
---

## [Editor Report · Acceptance letter]

3 Jun 2022

PONE-D-21-40491R1 

Suppression of choroidal neovascularization and epithelial-mesenchymal transition in retinal pigmented epithelium by adeno-associated virus-mediated overexpression of CCN5 in mice 

Dear Dr. Park:

I'm pleased to inform you that your manuscript has been deemed suitable for publication in PLOS ONE. Congratulations! Your manuscript is now with our production department. 

Kind regards, 

on behalf of

Dr. Ravirajsinh Jadeja 

Academic Editor

PLOS ONE